# Multiparametric MRI in Era of Artificial Intelligence for Bladder Cancer Therapies

**DOI:** 10.3390/cancers15225468

**Published:** 2023-11-18

**Authors:** Oguz Akin, Alfonso Lema-Dopico, Ramesh Paudyal, Amaresha Shridhar Konar, Thomas L. Chenevert, Dariya Malyarenko, Lubomir Hadjiiski, Hikmat Al-Ahmadie, Alvin C. Goh, Bernard Bochner, Jonathan Rosenberg, Lawrence H. Schwartz, Amita Shukla-Dave

**Affiliations:** 1Department of Radiology, Memorial Sloan Kettering Cancer Center, New York, NY 10065, USA; 2Department of Medical Physics, Memorial Sloan Kettering Cancer, New York, NY 10065, USA; 3Department of Radiology, University of Michigan, Ann Arbor, MI 48109, USA; 4Department of Pathology, Memorial Sloan Kettering Cancer Center, New York, NY 10065, USA; 5Department of Medicine, Memorial Sloan Kettering Cancer Center, New York, NY 10065, USA; 6Department of Surgery, Memorial Sloan Kettering Cancer Center, New York, NY 10065, USA

**Keywords:** bladder cancer, quantitative, multiparametric, artificial intelligence, diffusion-weighted MRI, dynamic contrast-enhanced MRI, radiomics

## Abstract

**Simple Summary:**

Bladder cancer is the sixth most common cancer in the United States. The prognosis is excellent for localized forms, but the survival rates drop significantly when cancer invades the smooth muscle of the bladder. Imaging is essential for the accurate staging, prognosis, and assessment of therapeutic efficacy in bladder cancer and has the potential to guide personalized treatment strategies. Computed tomography has traditionally been the standard modality, but magnetic resonance imaging (MRI) is the emerging technique of choice for its superior soft tissue contrast without exposure to ionizing radiation. Multiparametric (mp)MRI provides physiological data interrogating the biology of the tumor, as well as high-resolution anatomical images. Advanced MRI techniques have enabled new imaging-based clinical endpoints, including novel scoring systems for tumor staging. Artificial intelligence (AI) holds the potential for the automated discovery of clinically relevant patterns in mpMRI images of the bladder.

**Abstract:**

This review focuses on the principles, applications, and performance of mpMRI for bladder imaging. Quantitative imaging biomarkers (QIBs) derived from mpMRI are increasingly used in oncological applications, including tumor staging, prognosis, and assessment of treatment response. To standardize mpMRI acquisition and interpretation, an expert panel developed the Vesical Imaging–Reporting and Data System (VI-RADS). Many studies confirm the standardization and high degree of inter-reader agreement to discriminate muscle invasiveness in bladder cancer, supporting VI-RADS implementation in routine clinical practice. The standard MRI sequences for VI-RADS scoring are anatomical imaging, including T_2_w images, and physiological imaging with diffusion-weighted MRI (DW-MRI) and dynamic contrast-enhanced MRI (DCE-MRI). Physiological QIBs derived from analysis of DW- and DCE-MRI data and radiomic image features extracted from mpMRI images play an important role in bladder cancer. The current development of AI tools for analyzing mpMRI data and their potential impact on bladder imaging are surveyed. AI architectures are often implemented based on convolutional neural networks (CNNs), focusing on narrow/specific tasks. The application of AI can substantially impact bladder imaging clinical workflows; for example, manual tumor segmentation, which demands high time commitment and has inter-reader variability, can be replaced by an autosegmentation tool. The use of mpMRI and AI is projected to drive the field toward the personalized management of bladder cancer patients.

## 1. Introduction

Bladder cancer is the sixth most common cancer in the United States, with an incidence in men approximately four times higher than in women [1]. Tobacco smoking is implicated in about 50% of bladder cancer cases, followed by other environmental risk factors, such as exposure to chemicals and industrial pollutants [2]. Squamous cell carcinoma and adenocarcinoma are rare subtypes (incidences of 6–8% and 2%, respectively) [3]. While localized forms of bladder cancer have an excellent prognosis, survival rates drop significantly if the smooth muscle is invaded [4]. Accordingly, the key classification for bladder cancers is non-muscle-invasive bladder cancer (NMIBC) vs. muscle-invasive bladder cancer (MIBC) [5]. Cystoscopy is the standard procedure for diagnosing and treating bladder cancer, allowing direct access to a tumor for biopsy, fulguration, and/or resection. Radical cystectomy (RC) is the treatment option for MIBC [6]. However, some patients may not meet the inclusion criteria for RC due to morbidity and perioperative risks. Multimodality treatment approaches are evolving, including bladder-sparing in MIBC [7]. The advantages of bladder-sparing procedures include less aggressive surgery, the avoidance of urinary diversion, the preservation of sexual potency, and improved quality of life [8]. The standard treatment is transurethral resection for NMIBC and RC, and neoadjuvant chemotherapy (NAC) for MIBC [9]. Recently, the treatment landscape has been transformed by paradigm-changing breakthroughs with two new classes of drugs, antibody-drug conjugates (AbDCs) and immune checkpoint inhibitors (ICIs), which offer more effective and less toxic treatment options [10,11,12,13].

Medical imaging with computed tomography (CT) and magnetic resonance imaging (MRI) has been used to derive quantitative imaging biomarkers (QIBs) that provide oncologists with clinical endpoints for the staging, prognosis, and assessment of therapeutic efficacy [14]. Recent technical advances in multimodality imaging have impacted the management of patients with bladder cancer. While CT has long been the standard imaging modality for bladder cancer treatment evaluation, MRI is emerging as a modality of choice because of its superior soft tissue contrast without exposure to ionizing radiation [15,16]. MRI differentiates the bladder wall layers and enables an accurate assessment of the depth of tumor invasion and extravesical extension [17]. The advancement in MRI technology has led to the implementation of multiparametric (mp)MRI, which provides multi-*b*-value diffusion-weighted (DW) and dynamic contrast-enhanced (DCE) image sets. mpMRI acquires anatomic and physiological images for qualitative evaluation and allows for the measurement of model-based quantitative parameters [18,19]. The acquisition, interpretation, and reporting of mpMRI for bladder cancer were standardized by VI-RADS, the Vesical Imaging–Reporting and Data System, which was developed in 2018 through expert consensus [20,21]. VI-RADS’s scoring comprises T_2_-weighted (w), DW-, and DCE-MRI as a qualitative assessment for diagnostic reporting [22,23]. The dominant MRI sequence for bladder cancer risk assessment is DW. If DW-MRI is suboptimal, DCE-MRI is considered a second option.

Quantitative metrics derived from DW-, DCE-MRI, and radiomics enable numerical values that promise improvements in detecting, staging, and evaluating treatment response in bladder cancer [24,25,26]. DW-derived metrics include the apparent diffusion coefficient (ADC, the composite of both diffusion and capillary perfusion), apparent kurtosis coefficient (*K_app_*, surrogate of tissue microstructure), and true diffusion coefficient (D, capturing tumor cellularity, extracellular-space tortuosity, and integrity of cellular membranes) [27,28,29]. ADC metric values were able to identify the aggressive phenotype of upper urinary tract urothelial cell carcinoma and monitor response to chemo-radiotherapy [30] and NAC therapy in MIBC [31,32]. DCE-derived biomarkers such as plasma perfusion (*Fp*) and the volume transfer constant (*K^trans^*) are surrogates of tumor perfusion and permeability [19,33,34]. *Fp* values were able to distinguish between residual tumor and therapeutic effects in MIBC patients treated with NAC [19]. The entropy of *K^trans^*, a measure of heterogeneity, was significantly lower in responders than in nonresponders after NAC therapy in bladder cancer patients [35]. QIBs from DW- and DCE-MRI also allow clinicians to develop personalized treatment strategies [36].

Radiomics is an approach that quantifies textural information in mpMRI images by mathematically extracting the spatial distribution of signal intensities and pixel interrelationships [37]. Radiomics also includes morphological (shape, size, and volume) and gray-level features (contrast, etc.), in addition to the texture features [38]. Extracted radiomic features have been used to characterize MIBC and predict tumor response to NAC in MIBC patients [26,39,40].

New artificial intelligence (AI) data analysis methods offer promise in developing biomarkers through the automated discovery of radiomic features in qualitative and quantitative mpMRI data associated with clinical outcomes in the training dataset. This approach has the potential to go beyond existing biomarkers developed using traditional statistical methods, extracting and analyzing hidden information about individual patients that can guide the personalized management of their cancer [37,41]. This review discusses quantitative mpMRI acquisition for the bladder and the current and potential role of derived QIBs for clinical endpoints, including staging, prognosis, and prediction of treatment efficacy in MIBC [19,25,32,34]. Future directions for AI are highlighted, including the specific clinical tasks that are likely to benefit from narrow AI applications, along with the challenges that AI is expected to face and recommendations for its successful application in MIBC.

## 2. Multiparametric (mp)MRI and VI-RADS Score in Bladder Cancer

mpMRI increasingly surpasses CT as the preferred imaging technique for bladder cancer because of its superior spatial resolution and more reliable characterization of the bladder’s layers and locoregional anatomic structures [15,42]. mpMRI exhibited a high diagnostic performance in differentiating NMIBC from MIBC and predicting extravesical extension [42].

### 2.1. Qualitative mpMRI

High-resolution qualitative mpMRI images acquired using standard multiplanar imaging protocols are used for the preoperative local staging of bladder cancer, including fat-saturated T_2_w, precontrast T_1_w, and postcontrast T_1_w sequences [21]. T_1_w images identify extravesical fat infiltration, pelvic lymphadenopathy, and bone metastases [17]. Urine in the bladder has low signal intensity on T_1_w images, normal detrusor muscle and bladder tumors both have intermediate signal intensity, and the adjacent fat has high signal intensity. Therefore, T_1_w images are valuable in showing the luminal extension of the tumor, as well as perivesical fat infiltration [43]. T_2_w images are used to assess the whole pelvis anatomy, including the bladder and surrounding tissue, for tumor detection and morphology evaluation [44]. Tumor invasion into adjacent organs (prostate, uterus, and vagina) can also be better evaluated on T_2_w than T_1_w images.

The standard MRI sequences used to perform VI-RADS scoring in radiological practice are T_2_w and postcontrast T_1_w images for anatomical imaging, and DW- and DCE-MRI for physiological information [21]. This protocol reduces the potential for mismatch of the lesion between sequences, as such a mismatch could lead to the erroneous upstaging or downstaging of tumors in bladder cancer [20,23]. VI-RADS shows good sensitivity and specificity for determining MIBC; however, technical factors associated with MRI acquisition and cutoff scores must be considered [45,46]. The inclusion of DW-MRI protocols for tumor staging has been found to improve the specificity of bladder cancer detection [47]. An inflammatory change or fibrosis surrounding the tumor may mimic the invasion of bladder cancer on T_2_w or postcontrast T_1_w. To reduce the resulting potential for over-staging, DW-MRI was added to VI-RAD’s protocol to improve bladder cancer differentiation because benign formations would show no significant changes in signal intensity compared to tumors on DW images [21]. T_1_w acquisition is also helpful in diagnosing eventual hemorrhage and blood clots in the bladder and bone metastasis; however, its findings do not contribute to the score [48]. The postcontrast T_1_w images were useful for detecting early-stage disease. Numerous studies have highlighted the improved sensitivity, specificity, and accuracy when two or more sequences are used for diagnosing and staging bladder cancer [49,50].

VI-RADS has been consistently validated across several different institutions for the local staging of bladder cancer [51,52,53] and has been proven to contribute to diagnostic workup and disease management. VI-RADS criteria are used to calculate a score for each MRI sequence that represents the overall likelihood of cancer invading the muscle and beyond, from 1 (highly unlikely) to 5 (very likely), as presented in Table 1 [21]. The typical mpMRI acquisition parameters for bladder imaging are given in Table 2. T_1_w, T_2_w, DW-MRI, and postcontrast T_1_w images representative of those used in VI-RADS scoring are shown in Figure 1. The ADC map is derived from the DW images (*b* = 0 and 700 s/mm^2^) using a standard monoexponential model. 

### 2.2. Quantitative (q)MRI

Parametric maps derived from qMRI techniques display physiological properties [54] that can provide quantitative information and insights into tumor heterogeneity [55]. The most commonly used qMRI methods are DW- and DCE-MRI. DW-MRI quantifies ADC metrics that reflect the hindered motion of water molecules affected by tumor cellularity and tissue microstructure. DCE-MRI estimates the microvascular properties of tumor tissue by utilizing the kinetics of exogenous contrast agent (CA). A quantitative analysis of DCE images generates kinetic parameters that reflect tumor perfusion/permeability and CA distribution spaces [56]. An alternative to biophysical models of mpMRI is image analysis via radiomics, a data-driven approach to quantify visually imperceptible statistical features [57].

#### 2.2.1. DW-MRI

The DW-MRI acquisition used in the mpMRI protocol for VI-RADS scoring in bladder imaging images the tissue signal contrast affected by the Brownian motion of water molecules [58]. The cell membranes and intracellular organelles restrict or hinder the displacement of water molecules in tissue, causing lower attenuation (dephasing) of the MRI signal compared to free diffusion [59]. The amount of signal attenuation is evoked by the degree of diffusion weighting (*b*-value) determined by the amplitude, duration, and separation of applied diffusion gradient pulses [60]. Single-shot echo-planar imaging (SS-EPI) is the default method of choice in radiology clinics for performing DW-MRI. The typical acquisition parameters for the bladder are given in Table 2. Compared to other techniques, SS-EPI offers a shorter scan time, higher signal-to-noise ratio (SNR), and greater immunity to respiratory motion [61]. Its limitations include geometric distortion due to B_0_ inhomogeneity, signal dropout, and image blurring from field inhomogeneity and eddy currents [62]. The introduction of parallel imaging with a reduced echo time, increasing the number of excitations (NEX), and adjusting the matrix and voxel size can dramatically reduce distortion and blurring and maintain sufficient spatial resolution and desired SNR [63]. Recently, deep learning (DL) has been applied to the image reconstruction of DW sequences and has shown promise to enhance the quality of bladder imaging by reducing the scan time and improving image quality [64].

**Table 2 cancers-15-05468-t002:** mpMRI bladder imaging protocol for 1.5 and 3.0 Tesla (T).

Parameter	T_2_w	DW	DCE
Field strength	1.5 T/3 T	1.5 T/3 T	1.5 T/3 T
Sequence *	FSE	SS-EPI	FSPGR
Plane orientation	Multiplanar	Axial	Axial
FOV (mm)	220–250	250–300	250–300
TR (ms)	4000–5000	4500–6000	3.5–4.5
TE (ms)	80–120	60–80 (minimum)	1.2–2.2
Acquisition matrix	256 × 192–256	128 × 128	256 × 192–214
Slice thickness/gap (mm)	3–4/0	3–4/0	3–4/0
Number of excitations	1–2	4–12 ^+^	1
Flip angles (FAs) (degree)	90	90	15
*b*-values (s/mm^2^)	NA	0 and 800–1000, up to 2000 optional	NA

Note: Precontrast T_1_ mapping with multiple flip angles (30°, 15°, and 5°) is preferred; * fast spin echo (FSE), single-shot echo-planar imaging (SS-EPI), and fast spoiled gradient echo (FSPGR). ^+^ To improve SNR, more excitations are required at higher *b*-values

To fit the DW imaging data, a standard monoexponential model requires at least two *b*-values (*b* = 0 s/mm^2^ and a high *b*-value of 800–1000 s/mm^2^) to calculate the apparent diffusion coefficient (*ADC* (mm^2^/s); Equation (1) [60]).
(1)$$s_{b}=s_{0}\;e^{-b\times ADC}$$
where *S_b_* and *S*_0_ denote signal intensities with and without diffusion weighting, respectively; and *b* is the diffusion-sensitizing factor or *b*-value.

The monoexponential modeling of signal decay as a function of *b*-value for DW images assumes a Gaussian distribution for the displacement of water molecules over the given measurement interval [58]. LeBihan formulated the Gaussian-based intravoxel incoherent motion (IVIM) model (Equation (2)), which provides estimates of pseudo-diffusion (perfusion) phenomena of microcirculation of blood in the capillary network (at low *b*-values < 100 s/mm^2^) and molecular diffusion in tissue (intermediate *b*-values) [65]. The translation of a water molecule substantially deviates from a Gaussian to non-Gaussian behavior due to complex cellular structures of tissue being more readily probed with greater diffusion weighting (>1000 s/mm^2^) [66]. The second-order expansion of the signal decay at high *b*-values (>1000 s/mm^2^), accounting for the deviation from monoexponential behavior, is termed diffusion kurtosis imaging (DKI) (Equation (3)) [67].
(2)$$s_{b}=s_{0}\;[f\;e^{-b\times D^{*}}+(1-f)e^{-b\times D}]\;\;$$
(3)$$s_{b}=s_{0}\;e^{-b\times D_{app}+\frac{1}{6}K_{app}(b\times D_{app}{)}^{2}}$$

DKI’s apparent kurtosis coefficient, *K_app_* (unitless), quantifies the degree of diffusion restriction by heterogeneous subcellular structures. Its apparent diffusion coefficient, *D_app_* (mm^2^/s), characterizes the tissue microstructure. Mathematically, *K_app_* describes the “peakedness” of the distribution function, representing the deviation of the diffusion propagator from a normal (Gaussian) shape [67]. The advanced non-Gaussian (NG)-IVIM model incorporates the kurtosis coefficient, *K*, into the IVIM model [65,68]. The nested models, IVIM, DKI, and monoexponential, can be derived from the NG-IVIM model setting *K_app_* and/or *f* = 0 [68]. Standard DW-MRI [49,69,70] and DKI models [28,29] have been used for bladder cancer imaging.

#### 2.2.2. DCE-MRI

DCE-MRI is another component of the VI-RADS protocol [21]. It reveals qualitative changes in signal intensity as a function of time after CA injection, which is routinely used for diagnostic evaluation of blood perfusion kinetics in tissue. DCE data for bladder cancer can be analyzed semi-quantitatively and quantitatively. The semi-quantitative approach provides descriptive parameters, such as the time to peak (TTP), initial enhancement (*IE*), wash-in rate (WIR), wash-out rate (WOR), area under contrast curve (AUC), and maximum signal enhancement (SE) [71].

Quantitative DCE-MRI parameters allow for the spatial mapping of heterogeneous tumor characteristics for improved staging and treatment planning. The different portions of a signal intensity time-course curve are characterized; for example, the initial upslope, including the peak height, reflects the total blood flow (*F_p_*) and plasma volume fraction (*v_p_*). The downslope curve is due to contrast leakage into the extravascular extracellular space (EES), reflecting vascular permeability. The volume fraction of EES (*v_e_*) can be inferred from the late portion of the curve. The quantitative approach of the DCE data analysis uses pharmacokinetic models. It is performed through the following steps: (i) conversion of signal intensity to CA concentration through the longitudinal R_1_ relaxation rate (R_1_ = 1/T_1_) (Equation (2)), (ii) selection of an appropriate pharmacokinetic model (Equation (3)), and (iii) estimation of pharmacokinetic model parameters [72]. The model-derived quantitative parameters include the *F_p_*, vascular permeability surface areas product (PS), EES, etc. The extended Tofts model (ETM) has become a common approach for the pharmacokinetic modeling of DCE-MRI data. ETM provides three kinetic parameters: the volume transfer constant, *K^trans^*; *v_p_*; and *v_e_* [73]. The ETM model assumes a bidirectional exchange of CA between blood plasma and EES and that the water molecule exchange between the tissue compartments is effectively infinitely fast. The fast exchange limit (FXL) assumes that the change in *R*_1_(*t*), ${∆R}_{t}\left(t\right)$, is linearly proportional to the contrast agent tissue concentration (*C_t_*(*t*)) [73], and it is given by the following:(4)$$R_{t}\left(t\right)=R_{10}+r_{1}C_{t}\left(t\right)\to {∆R}_{t}\left(t\right)={R_{1}(t)-R}_{10}=r_{1}C_{t}(t)\;$$

The temporal evolution of *C_t_*(*t*) is given by the following [73]:(5)$$C_{t}(t)=K^{trans}\int_{o}^{t}e^{-k_{ep}\left(t-\tau \right)}c_{p}\left(\tau \right)+v_{p}C_{p}(t)\;$$
where *r_1_* [mM^−1^ s^−1^] is the longitudinal relaxivity of a [Gd] based CA; C_p_(t) and C_t_(t) are the CA concentrations in the plasma space and tissue, respectively; and *k_ep_* is the backward flux from EES to plasma space, respectively.

A variant model from FXL is the fast exchange regime (i.e., also called shutter speed model) that accounts for the rate of water exchange between the intracellular and EES across the cell membrane, providing estimates of the mean lifetime of intracellular water molecules and an inverse of the rate of water exchange (*τ_i_*), in addition to *K^trans^* and *v_e_* [74]. This is associated with the metabolic activity of a cell [75]. The FXR model is usually valid only when a large amount of CA extravasates into the EES (i.e., *v_p_* ≅ 0).

Figure 2 shows example overlay parametric maps for mpMRI metrics derived from DCE and DW imaging of bladder cancer: *K^trans^*, *IE*, signal enhancement ratio (*SER*), perfusion-suppressed ADC (*b* = 100, *b* = 800), *f*, and *K_app_*. The most viable tumor regions tend to have higher *K^trans^*, *IE*, and *K_app_* and lower ADC and *f*, while the mid-anterior bladder wall exhibits normal characteristics.

#### 2.2.3. Radiomics

Radiomics is a data-driven approach that is used to quantify visually imperceptible statistical features from qualitative and quantitative MRI that can provide their correlation with tumor physiology with no a priori information [57]. Extracting radiomic features from qualitative MRI (T_1_w/T_2_w) and quantitative (DW- and DCE-MRI) requires some preprocessing steps [76]. Radiomics is a quantitative imaging feature extraction method that provides more information about the grayscale patterns and inter-pixel relationships. Radiomics can be used for the extraction of the traditional first-order, second-order, and high-order statistical image texture features based on the gray-level co-occurrence matrix (GLCM) and gray-level run-length matrix (GLRLM) from MR images [38,77]. First-order texture statistics are based on the histogram that describes the distribution of voxel intensities within the image region defined by the mask by using basic metrics such as energy, entropy, skewness, kurtosis, etc. The second- and higher-order features provide information about the inter-voxel relationships within the image [78]. Several open-source software packages, like PyRadiomics [79], CERR [80], LifeX [81], IBEX [82], CaPTK or CGITA [83], RaCaT [84], and RodiomiX [85], have been developed for extracting image features. Few radiomics tools have undergone comprehensive methodological standardization with external validation [57]. The accurate automated delineation of regions of interest (ROIs), including the tumor and the normal wall region, will be essential for the radiomics analysis to contribute to bladder cancer diagnosis and prognosis. Figure 3 displays the workflow of the image segmentation and radiomic feature extraction of MR images.

## 3. mpMRI for Clinical Consideration

The VI-RADS scoring is a standardized approach to imaging and reporting bladder cancer with mpMRI, which has changed the paradigms of bladder cancer detection and characterization. Advances in MRI technology have shown great promise for improved local staging and detecting local recurrences after treatment in bladder cancer. Previous studies have illustrated that mpMRI QIBs assessing tumor heterogeneity have the potential to improve the diagnosis, characterization, and assessment of treatment response in MIBC [19,25,34,70]. Image features and mpMRI-derived physiological QIBs can be further employed to develop computational models using AI algorithms that may serve as a guidance tool for personalized treatment.

### 3.1. mpMRI for Staging, Characterization, and Prognosis in MIBC

The TNM (tumor–node–metastasis) staging system is used for bladder cancer staging [86]. mpMRI techniques improve the accuracy of detecting bladder cancer and have been used for histologic grading and TNM staging of bladder cancer [20,87,88]. For prognosis, accurate staging is necessary prior to treatment. Preoperative mpMRI may provide useful information regarding treatment response [89]. The clinical stage alone is unreliable for determining tumor extension beyond the bladder wall, showing significantly higher recurrence rates and worse survival than those with organ-confined tumors [90]. Therefore, distinguishing between organ-confined and non-organ-confined tumors is essential. The application of mpMRI VI-RADS to the diagnosis of MIBC has shown excellent results [91]. Green et al. reported that mpMRI exhibited >80% accuracy in distinguishing between NMIBC and MIBC [92], a key component of selecting the optimal treatment strategy. Tekes et al. reported an accuracy of 85% in differentiating NMIBC from MIBC using gadolinium-based postcontrast T_1_w images [90]. Johi et al. suggested that adding DW-MRI and the derived ADC metric value to T_2_w improves the accuracy of MRI in bladder cancer detection and staging; staging accuracy was better in T_2_w plus DW-MRI (83%) as compared to DW-MRI alone (77%) or T_2_w alone (75%) [93].

Radiomics-based signatures from mpMRI developed for precision diagnosis and treatment may serve as a novel and powerful tool in modern precision medicine, determining the extent of the invasion of bladder cancer and its locations [94]. An MRI-based radiomics study demonstrated that image features extracted from MR images can distinguish the tumor grade in bladder cancer and could enhance staging and support a decision-making process [76]. Shi and Xu et al. reported that histogram and GLCM feature analysis of T_2_w images exhibited significant differences between bladder cancer and the bladder wall [95,96]. Xi et al. reported that the textural features from DW images could reflect the difference between low- and high-grade bladder cancer, particularly GLCM features from ADC maps [97].

### 3.2. mpMRI for Prediction of Treatment Response in MIBC

Treatment decisions for bladder cancer patients are mainly based on the depth of bladder wall invasion by the tumor. RC is the accepted standard of care (SOC) for patients with MIBC [6]. Most patients with MIBC undergo an RC, cisplatin-based NAC, as this approach improves survival in this population. However, this disease management significantly impacts the quality of life (QOL), as RC affects continence, sexual function, fertility, and bowel function [98]. Predictive biomarkers are critical to identifying patients who will respond to NAC so that potential toxicities from cytotoxic chemotherapy can be limited in patients who are unlikely to derive benefit [99]. Emerging immunotherapy alternatives include the development of antibodies directly targeting tumor cells, ICI antibodies, and chimeric antigen receptor T-cell therapies [13]. ICI therapy has revolutionized the approach to treating metastatic disease in several cancers, including melanoma, non-small cell lung cancer, and renal cell carcinoma [100,101].

Locally advanced or metastatic bladder cancer patients are not eligible for first-line treatment with SOC combination cisplatin-based chemotherapy due to poor performance status, impaired renal function, and other comorbidities [102]. Recently, the antibody–drug conjugate pembrolizumab showed better antitumor activity than conventional carboplatin-based chemotherapy in untreated patients with cisplatin-ineligible locally advanced/metastatic urothelial cancer [13]. The initial studies support the safety of combining checkpoint inhibitor immunotherapy with chemo–radiation in MIBC [103,104]. New bladder-sparing treatment modalities combining ICIs simultaneously with chemoradiotherapy (CRT) have also shown promise for MIBC patients [105,106].

Yoshida et al. performed a feasibility study with mpMRI to assess the therapeutic response to induction CRT for MIBC [30]. They concluded that DW-MRI acquired for bladder cancer could help predict the pathologic complete response (pCR), allowing for more optimal patient selection in bladder-sparing protocols. Ahmed et al. reported that the DCE-MRI-derived wash-out rate parameter and ADC derived from DW-MRI could potentially predict pCR, and their combination enhances the identification of a complete response to NAC in MIBC [107]. A PURE-01 ICI clinical trial with neoadjuvant pembrolizumab assessed the pCR (ypT0ypN0), using mpMRI, and concluded that this method could be an option to develop bladder-sparing approaches in future studies [108].

Radiomic features can further quantify the spatial heterogeneity of mpMRI metrics and be useful in predicting treatment response to NAC in patients with MIBC, providing a decision-support tool for personalized management [99]. The mpMRI-based radiomics nomogram has shown the potential to be a noninvasive tool for quantitatively predicting tumor response to NAC in patients with MIBC [26]. A previous study reported that AI-based image feature analysis showed a high diagnostic performance in predicting MIBC [109]. Selected works from the radiomics literature on mpMRI data for bladder cancer are summarized in Table 3.

### 3.3. Artificial Intelligence in Bladder Cancer

Figure 4 shows an example of an AI workflow in bladder cancer imaging. AI-specific tasks have a role in each cycle of the workflow, from protocol selection and image acquisition to interpretation and, finally, use in clinical decision making. A summary of select works from the AI literature on bladder MRI is provided in Table 4. A common limitation of the current AI literature is single-institution studies with small sample sizes. Multisite studies using larger patient cohorts are essential to address issues, including the bias in AI algorithms. Federated learning techniques may help remove some of the barriers multisite studies face in combining patient data across institutions.

The precise segmentation of bladder walls and tumor regions is essential for noninvasive identification for grading and staging and avoiding partial volume errors in quantifying QIBs derived from mpMRI. In recent years, AI algorithms have been employed for automated tumor identification, staging, grading, bladder wall segmentation, prediction of recurrence, treatment response, and overall survival [115].

Bladder segmentation with DL techniques has been explored primarily using CT images [116,117,118,119,120,121]. The autosegmentation of bladder walls and tumors using mpMRI is challenging due to bladder shape variations; weak boundaries; diverse intensity and inhomogeneity in urine; and variability across the population, particularly in tumor appearance [122]. A few studies from Table 4 used various DL-based autosegmentation and denoising methods for T_2_w MR images [122,123,124,125,126,127]. Dolz et al. used a deep CNN model with progressive dilated convolutional modules to segment multiple regions in T_2_w images of 60 bladder cancer patients [123]. They achieved a higher level of accuracy, with a mean Dice similarity coefficient of 0.98, 0.84, and 0.69 for inner wall, outer wall, and tumor region segmentation, respectively. They concluded that the DL method had a better diagnostic performance, shorter processing time, and robust generalizability relative to radiologists using VI-RADS, thus indicating good potential for diagnosing MIBC. Moribata et al. performed an autosegmentation in bladder cancer on MRI using CNN and investigated the robustness of image features automatically extracted from ADC maps [128]. This study demonstrated that multicontrast MR images exhibit a higher segmentation performance than single-contrast models. Image features calculated from the automatic segmentation results showed high reproducibility for the first-order, shape-based, and higher-order features. Thus, AI technology has been recognized as a promising tool for MBIC imaging. However, its performance needs validation with independent datasets, and integrating AI into routine clinical practice remains challenging. AI systems often fail to generalize to local populations and imaging protocols, offer opaque reasoning, or demonstrate fragility in complex real-world conditions [129,130,131,132,133]. Hence, substantial work is still needed in this field for clinical translation.

**Table 4 cancers-15-05468-t004:** Summary of selected works on artificial intelligence from the literature on bladder cancer.

#	Application	Reference	Dataset	Methods	Conclusion
1	Segmentation	Dolz et al. (2018) [123]	60 patients (training 40, validation 5, test 15)	U-Net yields precise segmentation of bladder walls and tumors on T_2_w.	Higher accuracy than standard CNN, especially for tumors.
2		Li et al. (2020) [122]	1092 MR images	U-Net with priors is applied to segment bladder walls and tumors on T_2_w.	The method improved the accuracy of bladder wall segmentation.
3		Yu et al. (2022) [126]	245 patients (training 220, test 25)	Path augmentation U-Net segmentation for bladder walls and tumors on T_2_w.	It can precisely extract bladder structures, especially small tumors.
4		Coroamă et al. (2023) [127]	33 patients	A low-complexity 3D U-Net with less than five layers for segmentation of bladder walls and tumors on T_2_w.	System for automated diagnosis of bladder tumors that can lead to higher reporting accuracy.
5		Moribata et al. (2023) [128]	170 patients (training 140, test 30)	U-Net could segment bladder cancer, and robust high-order radiomics features were extracted from ADC maps.	The model performed accurate segmentation of bladder cancer, and the extracted radiomics exhibited high reproducibility.
6	Classification	Zou et al. (2022) [130]	468 patients	Inception V3, CNN on T_2_w, recognizes the position of bladder walls and tumors.	Reliable method that can be more focused on features from the surrounding area of the tumor.
7		Sevcenco et al. (2018) [131]	51 patients (training 36, test 15)	A multilayer perceptron with one hidden layer on ADC maps.	Classifier model combining the ADC values with clinical–pathological information can identify patients at high risk for survival.
8		Li et al. (2023) [133]	Multicenter cohort of 89 (121) patients (tumors), 61 (93) from center 1, and 28 (28) from center 2. Tumors for training 93, test 28	3D ResNet50 CNN on T_2_w as a multitask model exhibits good diagnostic performance in predicting MIBC.	The method was lesion-focused and more reliable for clinical decisions.
9	Denoising	Taguchi et al. (2021) [124]	68 patients	VI-RADS validation CNN, with denoising reconstruction on T_2_w, discriminates between NMIBC and MIBC.	Combining VI-RADS with denoising CNN might improve diagnostic accuracy.
10		Watanabe et al. (2022) [125]	163 patients	VI-RADS validation CNN with denoising reconstruction on T_2_w and DW- predicts accurate MIBC without using DCE-MRI.	It achieved a comparable predictive accuracy for MIBC to that of conventional VI-RADS.

## 4. Discussion

A precise assessment of bladder cancer is an essential step toward personalized patient treatment. Qualitative and quantitative mpMRI offers excellent contrast resolution for preoperative staging, local recurrence detection, and treatment response assessment for bladder cancer. The VI-RADS scoring system was developed to meet the consensus need for qualitative mpMRI to improve tumor staging and has proven to be a robust predictor for differentiating MIBC and NMIBC [21,23]. Many studies confirm that the standardization and high degree of inter-reader agreement to discriminate NMIBC from MIBC supports the VI-RADS implementation in routine clinical practice for bladder imaging [20].

The quantitative numerical values for ADC maps, reflecting tumor cellularity, provided by DW-MRI have shown promise for clinical application in bladder cancer [134,135,136,137]. The advanced DW-MRI acquisition protocols can include high *b*-values, offering the ability to quantify the DKI-derived metrics interrogating tissue microstructure, in addition to ADC, which has shown promise in differentiating MIBC from NMIBC [69]. DCE-MRI-derived metrics are surrogates of tumor perfusion and permeability and have shown encouraging results for assessing tumor responses to treatment in MIBC [19,34]. The emerging role of radiomics in bladder cancer offers distinct morphologic characteristics and more insights into tumor heterogeneity [77]. Integrating radiomics with VI-RADS improved clinical performance compared to VI-RADS alone [138].

AI-based methods are revolutionizing routine diagnostic procedures [139]. Manual segmentation is time-consuming and subject to high inter-reader variability, making this common clinical task one of the first areas in which AI can contribute to imaging data analysis. Different AI architectures have been proposed to tackle the difficult task of separating the inner and outer bladder walls and the bladder from the background with MR images (Table 4). Some bladder regions are more challenging to segment because of the thicker appearance of the walls. Image features were included in a newly designed CNN model, yielding excellent reproducibility and reliability [140]. AI developments focused on narrow tasks using mpMR images will allow researchers to identify new AI-QIBs for clinical endpoints in bladder cancer. Multisite studies using larger patient cohorts are needed to increase AI training sets, improve generalizability, reduce bias, and enable clinical translation.

## 5. Conclusions

mpMRI-derived physiological biomarkers and radiomic features are important in bladder cancer clinical applications. Most recently, AI is starting to be applied to bladder cancer, and the first results suggest that it may offer a significant advantage for staging and grading. New AI methods are expected to appear with a significant impact on diagnosis tools and therapeutic protocols. Clinical studies and trials may benefit from new AI methods, dramatically improving the patient’s quality of life.

## Figures and Tables

**Figure 1 cancers-15-05468-f001:**
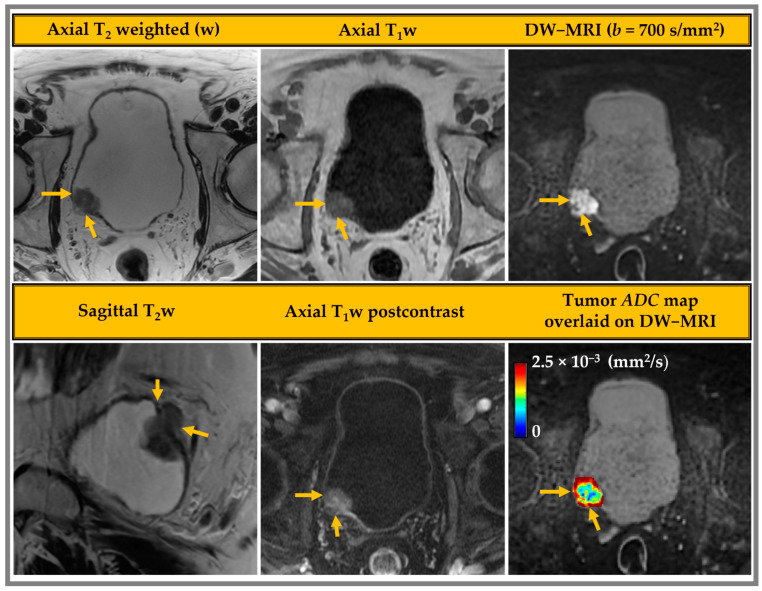
mpMRI images acquired from a 74-year-old male patient with bladder cancer. The multicontrast images include T_1_w, T_2_w, postcontrast T_1_w, and the diffusion-weighted magnetic resonance (DW-MR) image *b* = 700 s/mm^2^, with the apparent diffusion coefficient map overlaid. The yellow arrow points to a tumor with a VI-RADS score of 5.

**Figure 2 cancers-15-05468-f002:**
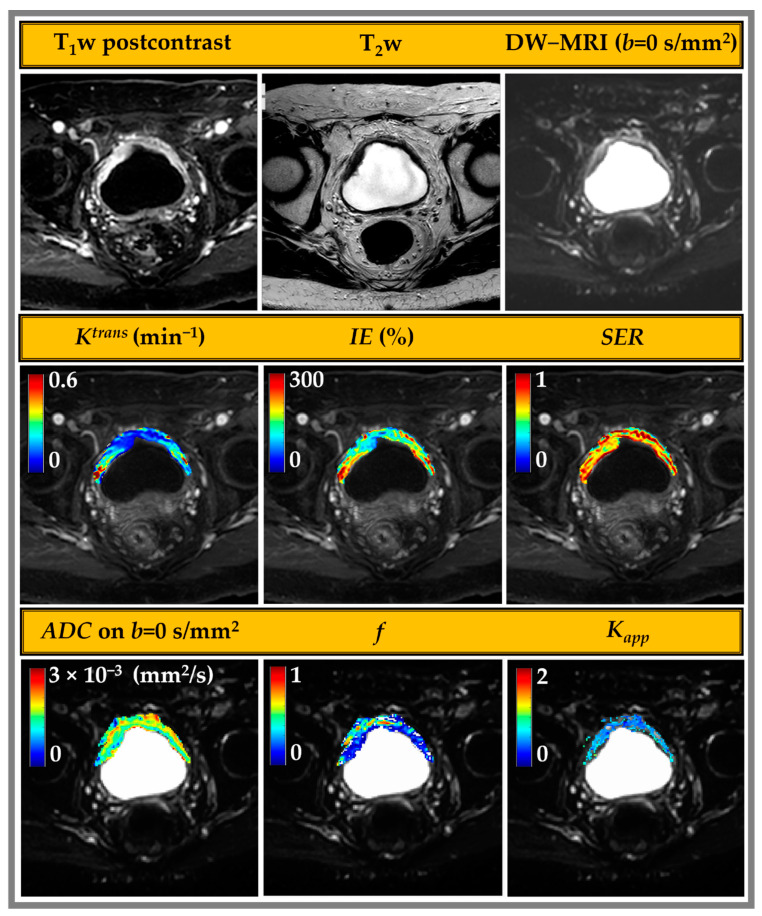
mpMRI images acquired for a 72-year-old male patient with bladder cancer. The top row shows the single slice images of the bladder (**left** to **right**): delayed-phase T_1_w (20-phase DCE-MRI, 7s sampling), T_2_w, and *b*-value = 0 s/mm^2^ DW-MR image. The middle row shows tumor overlays of volume transfer constant (*K^trans^*), initial enhancement (*IE*)%, and signal enhancement ratio (*SER*) derived from DCE. The lower row shows the overlay maps for DW-MRI-derived metrics (**left** to **right**): perfusion-suppressed apparent diffusion coefficient (ADC (*b* = 100, *b* = 800)), perfusion fraction (*f*), and apparent Kurtosis coefficient (*K_app_*).

**Figure 3 cancers-15-05468-f003:**
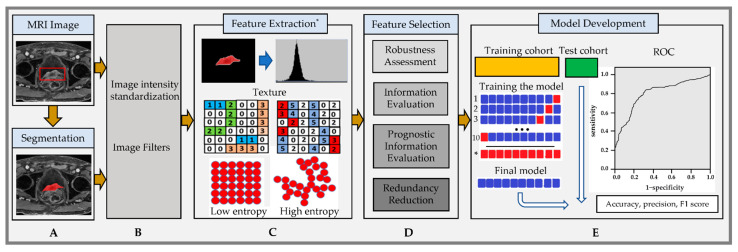
Radiomic feature analysis workflow: (**A**) Segmentation of tumor (bounding box in red) on postcontrast T_1_w images acquired from an 82-year-old male patient with bladder cancer extending to the prostate with a VIRADS score of 5. (**B**) Image standardization, a preprocessing step. (**C**) Feature extraction* from the region of interest with the histogram and textural image feature representation with low and high entropy. (**D**) Feature selection approaches. (**E**) Model development. Note: ** Extracted features should be compatible with the Image Biomarker Standardization Initiative (IBSI)* [57].

**Figure 4 cancers-15-05468-f004:**
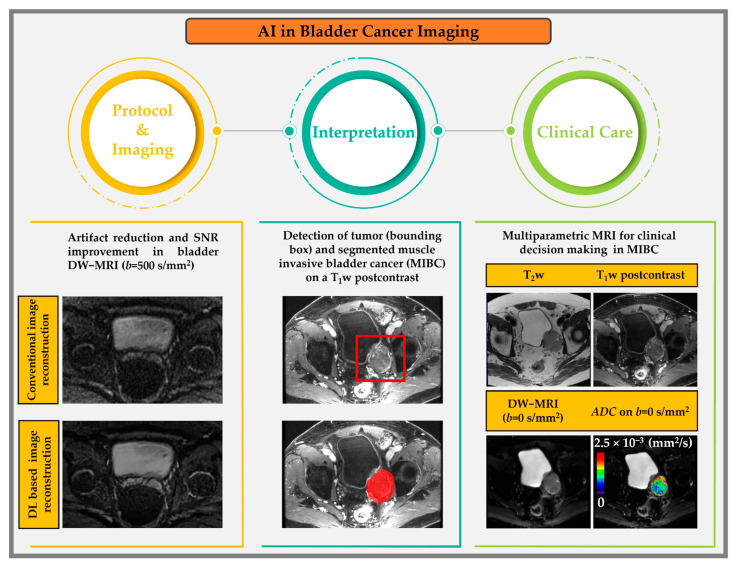
Artificial intelligence in the clinical radiology workflow, with examples from bladder MR imaging on a 44-year-old volunteer and a 73-year-old bladder cancer patient.

**Table 1 cancers-15-05468-t001:** VI-RADS score inference.

VI-RADS Score	Inferences
1	Muscle invasion is highly unlikely
2	Muscle invasion is unlikely to be present
3	Presence of muscle invasion is equivocal
4	Muscle invasion is likely
5	Invasion of muscle and beyond the bladder is very likely

**Table 3 cancers-15-05468-t003:** Summary of selected works from the radiomics literature on bladder cancer.

#	Reference	MRI/Segmentation/ Tools/Statistical Method	Dataset	Conclusion
1	Li et al. (2023) [110]	T_2_w and DW-MRI, manual, PyRadiomics, LASSO	3148 features, first order, shape and size, texture, wavelet filter, and Laplacian of Gaussian filter in 169 patients (70% training, 30% test); 24 optimal features	Radiomics combined with monograms can differentiate low-from high-grade NMIBCs.
2	Zhang et al. (2022) [26]	T_2_w, DW- and DCE-MRI, manual, and PyRadiomics	23,688 features, first order, shape, and grey levels (GLCM, GLRLM, GLSZM, GLDM, and NGTDM) in 342 patients (239 training, 68 validation); 43 optimal features	T_2_w, DW-MRI, and DCE-MRI radiomics models could effectively assess the state of muscular invasion.
3	Wang et al. (2020) [111]	T_2_w and DW-MRI, manual, LASSO, logistic regression, and SVM-RFE	1404 features, histogram, co-occurrence matrices, run-length matrix, and grey levels (NGTDM and GLRSZM) in 106 patients (64 training, 42 validation), 36 optimal features	Features selected by SVM-RFE reflect the regional heterogeneity of tumor tissues and can better characterize tissue heterogeneity differences between NMIBC and MIBC.
4	Xu et al. (2019) [112]	T_2_w, DW- and DCE-MRI, manual, SVM-RFE and LASSO	1872 features, histogram, co-occurrence matrices, run-length matrix, and grey levels (NGTDM and GLSZM) in 71 patients (50 training, 21 validation), 24 optimal features	The radiomics–clinical nomogram has potential in the preoperative prediction of the first two years after transurethral resection of the bladder tumor.
5	Zheng et al. (2021) [113]	T_2_w and DCE-MRI, manual, PyRadiomics, and SMOTE-LASSO	2436 features, 179 patients (70% training, 30% validation), 10 optimal features	The applied model could predict the Ki67 expression status and was associated with survival outcomes.
6	Kimura et al. 2022 [114]	ADC maps, manual and LIFEx, LIFEX, RF, and SVM	46 features: histogram, shape, grey levels (GLCM, GLRLM, GLZLM, and NGLDM) in 45 patients,	The radiomics model can predict the CRT response and serve as a novel imaging biomarker.

Note: gray-level co-occurrence matrix (GLCM), gray-level dependence matrix (GLDM), gray-level run-length matrix (GLRLM), gray-level size-zone matrix (GLSZM), Least Absolute Shrinkage and Selection Operator (LASSO), neighborhood gray-tone difference matrix (NGTDM), synthetic minority oversampling technique (SMOTE), support vector machines (SVMs), and Recursive Feature Elimination (RFE).

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
