# Peer review of "Multiparametric MRI in Era of Artificial Intelligence for Bladder Cancer Therapies"

_cancers, 2023, doi:10.3390/cancers15225468_

Round 1

Reviewer 1 Report

Comments and Suggestions for Authors

A detailed narrative review summarising the use areas of multi parametric MRI (mpMRI) for bladder cancer (BCa) with additional help of artificial intelligence (AI).

I have some minor comments that are listed below:

1. The manuscript was written in good English. 

2. It would have been better if the lines have been numbered, which would facilitate to locate comments. Luckily, very few comments have been raised to this manuscript.

3. A repeat of open terms for some of the abbreviations have been used (i.e., contrast agent, radical cystectomy, immune checkpoint inhibitor, etc.). This can be checked. Having said about abbreviations, I think BCa or BC or BLC can be used as an abbreviation for bladder cancer, which has been used numerously times throughout the manuscript. Additionally, even that may not be a journal requirement, as there are a lot of abbreviations in this manuscript, I suggest the authors to add a list of abbreviations with their meanings at the end of the main text.

4. The asterisk signs (* and **) should be explained as a footnote in Table 2 among the abbreviations (TR, TE, FRFSE, etc.) used. Additionally, I think it would be T1 mapping instead of T10 (written in the note on page 7).

5. Figure 1 is missing or the name of Figure 2 would be 1. Moreover, I think it might be better if the figures are printed a little bigger.

Author Response

Reviewer 1 (R1) Comments and Suggestions for Authors

I have some minor comments that are listed below:

Response: We greatly appreciate your thorough reading of our invited review and providing valuable comments to improve the quality of the manuscript for a special Cancers issue: "Methods and Technologies Development." We have carefully considered the comments and thoroughly revised the manuscript to address them. In the revised manuscript, we have incorporated the comments and suggestions with track changes (reviewers' comments are in italics) and numbers (#). The author's edits are noted as "A" in the revised manuscript.

Below is a point-by-point response to the comments and suggestions.

R1. The manuscript was written in good English. 

Response: Thank you for your appreciation.

R2. It would have been better if the lines have been numbered, which would facilitate to locate comments. Luckily, very few comments have been raised to this manuscript.

Response: Thank you for pointing out this. The revised Word document and PDF have the line numbers on each page.

R3. A repeat of open terms for some of the abbreviations have been used (i.e., contrast agent, radical cystectomy, immune checkpoint inhibitor, etc.). This can be checked. Having said about abbreviations, I think BCa or BC or BLC can be used as an abbreviation for bladder cancer, which has been used numerously times throughout the manuscript. Additionally, even that may not be a journal requirement, as there are a lot of abbreviations in this manuscript, I suggest the authors to add a list of abbreviations with their meanings at the end of the main text.

Response:  We greatly appreciate your suggestion. We have added an abbreviation list at the end of the document.

R4. The asterisk signs (* and **) should be explained as a footnote in Table 2 among the abbreviations (TR, TE, FRFSE, etc.) used. Additionally, I think it would be T1 mapping instead of T10 (written in the note on page 7).

Response: Thank you for this helpful suggestion. The abbreviations are defined in Table 2 as a footnote.

R5. Figure 1 is missing, or the name of Figure 2 would be 1. Moreover, I think it might be better if the figures are printed a little bigger.

Response: Thank you for pointing it out. The revised Word document and PDF have the missing Figure 1.

Reviewer 2 Report

Comments and Suggestions for Authors

This paper was written to focus on the application, and performance of mp-MRI for Bladder imaging specifically the use of VI-RADS.

1.     The introduction is well drafted, just few minor points on articles cited, there are 48 articles cited in the introduction, can this be condensed, and only relevant articles can be cited.

2.     Radiomic features, it would be helpful to provide the steps and for getting the radiomic features (standard procedure) from the radiomic toolbox. Also, to mention which radiomic package would be preferred and the list of parameters (and values) used by the toolbox mpMRI to extract the Radiomic feature.

3.     Also, It would be helpful to provide the list of features that are extracted from radiomic features and have been found to be effective in predictive analysis. Summarizing articles that have shown improved predictions using only radiomic features with mpMRI (list of important features) and have greater prediction score.

4.     The datasets used by AI algorithms, it would be helpful to provide a table with data used by the articles cited, to know if the dataset was in-house and the size of training and test set. This would help in concluding the results and its impact on the overall procedure.

5.     Also, the Table2. is very broad ranging from segmentation, classification, and de-noising. It would be good to create separate table or mention articles related to the specific topic related to the main idea mention in the paper.

6.     Also, since you mention the AI based segmentation for specific regions in bladder cancer, it would be good to compare the results and to identify the best approach (current state of art) used by AI algorithms for the task of bladder cancer detection.

Comments on the Quality of English Language

English language used was good, and require no improvement.

Author Response

                Reviewer 2 (R2) Comments and Suggestions for Authors

We greatly appreciate your thorough reading of our invited review and providing valuable comments to improve the quality of the manuscript for a special Cancers issue: "Methods and Technologies Development." We have carefully considered the comments and thoroughly revised the manuscript to address them. We have incorporated the comments and suggestions with track changes (reviewers' comments are in italics) and numbers (#) in the revised manuscript. The author's edits are noted as "A#" in the revised manuscript.

  1. The introduction is well drafted, just few minor points on articles cited, there are 48 articles cited in the introduction, can this be condensed, and only relevant articles can be cited.

Response: Thank you for this helpful comment. We have reduced the number and cited only the relevant articles in the revised manuscript introduction.

  1. Radiomic features, it would be helpful to provide the steps and for getting the radiomic features (standard procedure) from the radiomic toolbox. Also, to mention which radiomic package would be preferred and the list of parameters (and values) used by the toolbox mpMRI to extract the Radiomic feature.

  Response: Thank you for this helpful comment. We have modified our radiomic figure 3 to exhibit the workflow for image feature extraction in the revised review article. We have also cited a few relevant references to those who have developed radiomic packages. The list of parameters (and values) extracted by these toolboxes is beyond the scope of the review article.

  1. Also, It would be helpful to provide the list of features that are extracted from radiomic features and have been found to be effective in predictive analysis. Summarizing articles that have shown improved predictions using only radiomic features with mpMRI (list of important features) and have greater prediction score.

Response: Thank you for this valuable comment, which helps us improve the review's clarity. We have added a new Table (#3) with recent relevant articles summarizing the list of radiomic features. Adding a detailed list of the important features in a Table is beyond the scope of this review article.

  1. The datasets used by AI algorithms, it would be helpful to provide a table with data used by the articles cited, to know if the dataset was in-house and the size of training and test set. This would help in concluding the results and its impact on the overall procedure.

  Response: Thank you for this helpful comment. We have added the relevant text with the details mentioned in the revised Table.

  1. Also, the Table 2. is very broad ranging from segmentation, classification, and de-noising. It would be good to create separate Table or mention articles related to the specific topic related to the main idea mention in the paper.

    Response: Thank you for pointing it out. We have reorganized and edited Table #4 (Table #3) accordingly in the revised manuscript.

  1. Also, since you mention the AI based segmentation for specific regions in bladder cancer, it would be good to compare the results and to identify the best approach (current state of art) used by AI algorithms for the task of bladder cancer detection.

Response: Thank you for this valuable comment; we added relevant text to the revised review.

Reviewer 3 Report

Comments and Suggestions for Authors

This is very comprehensive and contemporary review of very clinically significant subject: assessment of therapeutic efficiency, staging and diagnostic of Bladder cancer with non-invasive methods. The critical advantage of the current study is the multi-center nature and development of algorithm of using AI for an objective assessment of the disease. The review to my opinion might be published in the "Cancers".

Author Response

Reviewer 3(R3) Comments and Suggestions for Authors

This is very comprehensive and contemporary review of very clinically significant subject: assessment of therapeutic efficiency, staging and diagnostic of Bladder cancer with non-invasive methods. The critical advantage of the current study is the multi-center nature and development of algorithm of using AI for an objective assessment of the disease. The review to my opinion might be published in the "Cancers".

Response: We greatly appreciate your time and efforts in thoroughly reading our invited review paper for a special Cancers issue: "Methods and Technologies Development." We are happy to learn that there are no further comments to improve the quality of this manuscript. The author's edits are noted as "A#" in the revised manuscript. Thank you again for your consideration.
